# GraphULM: A Multi-Resolution CNN and GCN Framework for Ultrasound Localization Microscopy

**Mohammad Sabih**[*1]                                                              MVS7196@PSU.EDU
**Mohamed Khaled Almekkawy**[*1]                                                   MKA9@PSU.EDU
[1] *The Pennsylvania State University, School of Electrical Engineering and Computer Science, University Park, PA*

**Editors:** Accepted for publication at MIDL 2026

## Abstract

Ultrasound Localization Microscopy (ULM) is a prominent technique in medical imaging, widely applied to enhance super-resolution, particularly in in-vivo settings. The process of localization, followed by tracking of microbubble (MB), poses a significant challenge in ULM due to its intricacy and complexity. High MB densities intensify these challenges, thereby diminishing the performance of traditional methods and certain deep learning algorithms in achieving precise localization. We present GraphULM, a novel and computationally efficient architecture that combines a Multi-Resolution Convolutional Neural Network (MRCNN) with a Graph Convolutional Network (GCN) to enhance localization efficacy in ULM. To develop an optimal training dataset, synthetically generated data is pre-combined with in-vivo b-mode samples, which improves feature diversity and generalization. Experimental evaluations in in-vivo demonstrate the model's high performance, reporting a localization precision of 21.9 µm, and a Jaccard index of 0.75, at a MB density of 2 MB/mm$^2$, underscoring the model's robustness. Additionally, our Frequency Ring Correlation (FRC) analysis reveals a remarkable resolution of 5.62 µm. The model operates at three times the speed of traditional pipelines, establishing its suitability for rapid ULM applications.

**Keywords:** Ultrasound Localization Microscopy, Multi-Resolution CNN, Graph Convolutional Network, Attention Mechanism, Medical Imaging

## 1. Introduction

Ultrasound Localization Microscopy (ULM) (Dencks and Schmitz, 2023) enhances the resolution of b-mode ultrasound and supports early disease detection (Denis et al., 2023) in cardiovascular and cerebrovascular applications (Demené et al., 2021). By leveraging optical super-resolution principles, MB contrast agents act as acoustic point scatterers that enable subwavelength imaging. ULM performance depends on MB localization, tracking, and vessel reconstruction, with motion correction and noise suppression improving stability (Heiles et al., 2022).

A major limitation of ULM is the long acquisition time (Hingot et al., 2019), driven by the trade-off between MB concentration and localization accuracy (Song et al., 2023). Low concentrations avoid PSF overlap but slow vessel coverage, whereas higher concentrations

---

* Contributed equally

increase ambiguity between adjacent MBs. Improving localization at high MB concentrations remains challenging (Shin et al., 2024). Traditional algorithms perform well when MBs are well separated (Couture et al., 2018; Heiles et al., 2022), and several methods attempt to handle overlapping MBs using spatiotemporal flow dynamics (Huang et al., 2020). Other approaches such as Fast-AWSALM (Zhang et al., 2019) address acquisition constraints.

Deep learning has been widely applied to ULM. The mSPCN-ULM framework (Liu et al., 2020) reduces processing time while improving localization in high-density settings. Swin-transformer–based models (Liu and Almekkawy, 2023a) further improve precision, and YOLO-based localization (Liu and Almekkawy, 2023b) increases efficiency, though grid limitations affect performance in dense regions. A 3D CNN has been used to learn temporal stacks (Brown and Hoyt, 2019), while an LSTM model maps spatiotemporal frames directly to velocity fields (Chen et al., 2023). RF-domain learning has also been explored (Hahne et al., 2024), though RF data and associated transformations remain complex.

Graph Convolutional Networks (GCNs) (Ding et al., 2022; Singh et al., 2023) capture relationships between image regions more effectively than CNNs. GCNs generalize convolution to irregular domains and can model asymmetric MB distributions, making them suitable for ULM. Prior work in medical imaging (Yan et al., 2019; Ravinder et al., 2023) shows their capability in analyzing complex structures, highlighting their relevance for distinguishing true MB signals from noise.

Motivated by these observations, we propose an architecture combining a Multi-Resolution CNN (MRCNN) and a Graph Convolution Network (GCN) to enhance ULM localization accuracy. The network includes GCN, successive MRCNN blocks, and an upsampling stage to generate a super-resolution output. In-vivo and synthetic datasets were used for evaluation, and the method demonstrates high resolution with low inference time.

## 2. Materials and Methods

The ULM pipeline involves RF data acquisition, beamforming to produce b-mode frames, and MB localization in the resulting images. After localization, tracking is required to recover microvascular structures. For this, we employ the Hungarian algorithm (Tang et al., 2020), which efficiently determines optimal associations between detections across frames. The following subsections outline each stage of the ULM workflow in detail.

### 2.1. Simulation Data

To train the network, approximately 10,000 synthetic b-mode images of MBs were generated to mimic in-vivo intensity and PSF characteristics. To estimate realistic PSF statistics, about 500 MBs were extracted from in-vivo frames and fitted with an anisotropic Gaussian, yielding lateral and axial standard deviation ranges of [0.3, 0.5] and [0.4, 0.7], respectively. These parameters were used to construct PSF kernels via the `mvnpdf` function in MATLAB. MBs were randomly positioned on a $256 \times 256$ grid and convolved with the PSF kernel (Equation (1)),

$$S_i(x, z) = I_i(x, z) \otimes PSF, \tag{1}$$

then downsampled to $32 \times 32$, corresponding to a pixel size of 0.1 mm. Gaussian noise with SNR sampled uniformly between 8–30 dB (sampled per frame) was added. The resulting

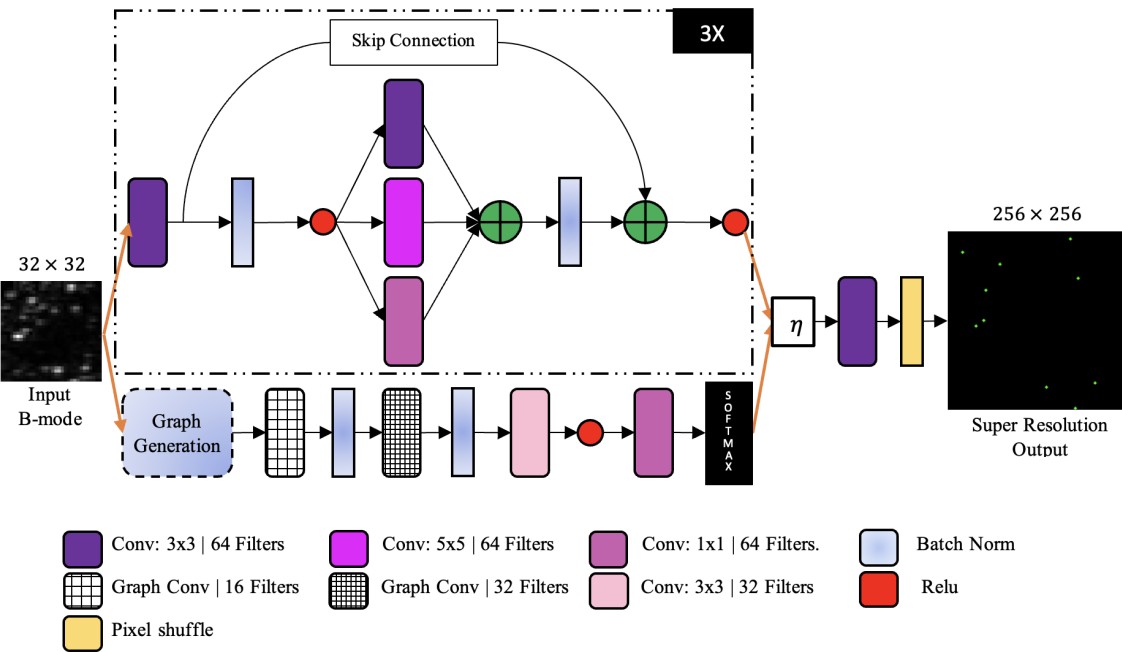

Figure 1: Network Architecture. The input b-mode is processed in tandem in two threads: the MRCNN and the GCN thread, which are later combined parametrically to produce a final super-resolution output of dimension $256 \times 256$.

dataset consists of $32 \times 32$ b-mode inputs and $256 \times 256$ target images, forming an upscaling factor of 8 and an effective output pixel size of 12.5 $\mu$m.

To improve generalization, 200 in-vivo b-mode frames were added to the training set. These frames were exclusive from the evaluation dataset. Their pseudo–ground truth was obtained using Radial Symmetry followed by manual selection to ensure high-quality samples.

## 2.2. Network Architecture

Figure 1 shows the architecture of the proposed network. Each b-mode frame is processed through two parallel threads, an MRCNN and a GCN, whose outputs are fused parametrically for precise localization. The MRCNN preserves the $32 \times 32$ input resolution until upsampling. Its first layer uses $3 \times 3$ filters (64 channels), followed by multiresolution processing with $1 \times 1$, $3 \times 3$, and $5 \times 5$ filters in parallel. These feature maps are fused, and this block is repeated three times with residual connections (He et al., 2016) to improve feature propagation and mitigate vanishing gradients. The MRCNN component employs a multi-resolution design with parallel filters of different sizes, a well-established technique in computer vision for multi-scale feature extraction.

In parallel, the GCN models the structural context of the b-mode frame. All pixels are treated as nodes, and edges connect adjacent pixels with weights based on intensity differences. The fundamental operation of graph convolution can be visualized using Equation (2).

$$H^{(l+1)} = \sigma \left( D^{-\frac{1}{2}} A D^{\frac{1}{2}} H^{(l)} W^{(l)} \right) \tag{2}$$

In the above equation, $H^l$ and $W^l$ are the processed feature matrix and the weight matrix on layer $l$ respectively, $A$ is the adjacency matrix, $D$ is the degree matrix of $A$ and $\sigma$ is the activation function; ReLU.

Two GCN layers with 16 and 32 filters are applied, followed by two convolutional layers, the latter using softmax activation to impose soft attention (Shen et al., 2018). The GCN output is fused with the MRCNN features using, Equation (3), where $\eta$ is learnable.

$$\text{Combination} = (1 - \eta) \cdot \text{MRCNN} + \eta \cdot (\text{MRCNN} \cdot \text{GCN}), \tag{3}$$

The fused representation is passed through the final layers, including pixel shuffle, to generate a $256 \times 256$ super-resolution output. The MRCNN captures fine PSF details through parallel multi-resolution filters ($1 \times 1$, $3 \times 3$, $5 \times 5$), enabling multi-receptive-field feature learning. The GCN operates in tandem to model global structural context by learning a B-mode graph with intensity-defined edge weights, allowing the network to leverage spatial relationships across the entire image. Together, these components are complementary, particularly in high-density scenarios, as demonstrated by the ablation experiments. We refer to the complete network as GraphULM.

## 2.3. Training Strategy

For training, since MBs are sparse, direct MSE (Hodson, 2022) can be unstable; thus, we compute MSE between the predicted and ground-truth maps after convolution with a $9 \times 9$ Gaussian kernel (G), as shown in Equation (4). Here, $\hat{y}$ and $y$ represent the predicted and target labels, respectively, for each sample $i$.

$$\text{Loss} = \frac{1}{N} \sum_{i=1}^{N} (\hat{y} \otimes G - y \otimes G)^2. \tag{4}$$

The fusion parameter $\eta$ is initialized to 0.5 so both network threads receive equal weight at the start of training. Optimization is performed using ADAM (Zhang, 2018) with an initial learning rate of 0.001, decayed by a factor of 0.05 every 10 epochs. This slightly aggressive schedule improves convergence and generalization for the ULM localization task.

## 3. Results and Discussions

During the design phase, it became evident that the MRCNN and GCN threads captured complementary information. The former models fine PSF details, and the latter captures the global structural context. This motivated the development of a unified architecture in which both pathways contribute to the final representation. Following training, parameter $\eta$ converged to value of 0.7. This shift from 0.5 to 0.7 suggests that the GCN component

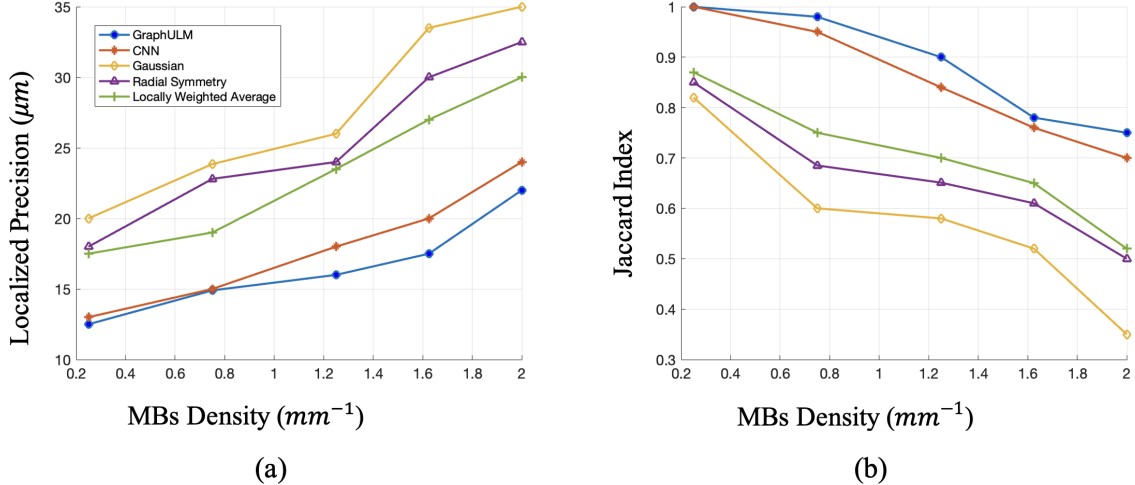

Figure 2: Performance comparison of GraphULM and four other methods in terms of (a) Localized Precision and (b) Jaccard index across varying MB densities. Compared to other methods, GraphULM demonstrates superior localization accuracy, particularly at higher densities.

contributed significantly towards improving localization accuracy, thus receiving greater weight in the combined model. We observed the same convergence behavior even when $\eta$ was initialized marginally higher or lower than 0.5, suggesting a stable learning dynamic. This section discusses performance on widely employed metrics and methods used in ULM experiments, defined in APPENDIX and (Liu and Almekkawy, 2023a; Hingot et al., 2021).

### 3.1. Reported Evaluation on the synthetic data

To evaluate localization performance under realistic noise conditions, a synthetic dataset of 100 frames ($32 \times 32$) was generated with varying MB densities and SNR levels between 8–30 dB, matching in-vivo characteristics. Figure 2(a) and Figure 2(b) show localized precision and Jaccard index for all methods. The MB count–density pairs used were $\{(5, 0.4), (10, 0.8), (15, 1.2), (20, 1.6), (25, 2)\}$.

In Figure 2(a), both deep-learning methods perform well across densities, with increased errors at higher densities due to overlapping PSFs. GraphULM outperforms CNN (Liu et al., 2020) in these conditions, achieving a precision of 21.9 $\mu$m at 2 MB/mm². This superior performance can be attributed to its ability to learn the context of all regions across the image at multiple scales, enabling it to analyze a broader picture and making better predictions.

From Figure 2(b), we can infer that all methods performed comparably well, achieving a Jaccard index greater than 0.8 in low MB density scenarios. This result is expected because, under low-density conditions, the MB count is low, leading to well-spaced PSFs over the b-mode image. Consequently, most predictions are likely to be true positives, and

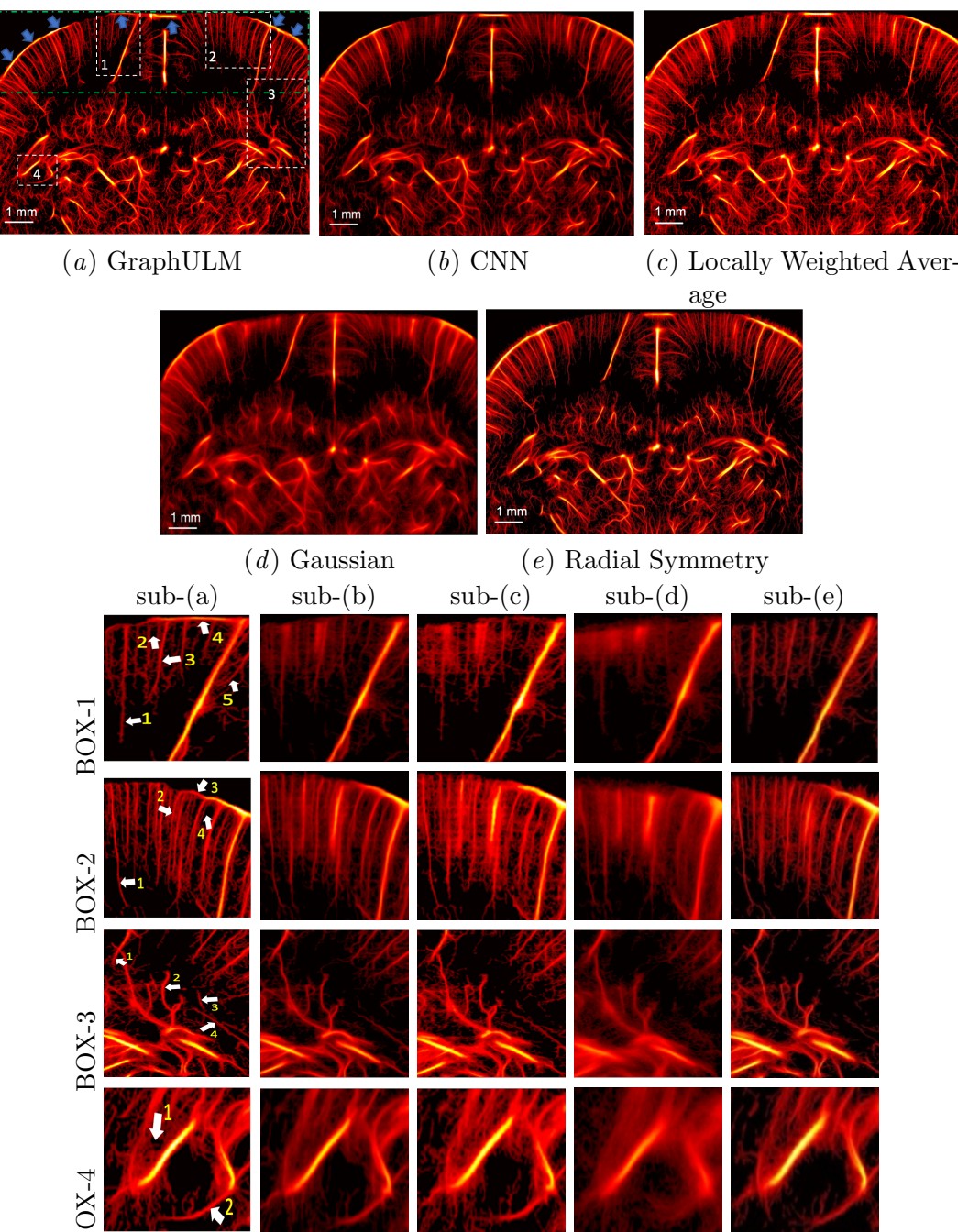

Figure 3: In vivo super-resolution images were obtained using five methods: (a) GraphULM, (b) CNN, (c) Locally Weighted Average, (d) Gaussian, and (e) Radial Symmetry. To facilitate a detailed comparative analysis, zoomed-in views of four specific regions (labeled BOX-1 to BOX-4) are provided in the lower rows, corresponding to the regions of interest marked in (a). These zoomed-in images highlight subtle differences in resolution and structural clarity between the methods, enabling a closer evaluation of the ability of each approach to capture fine vascular details.

unless the noise is anomalous, the probability of false positives remains very low. Furthermore, we can deduce that both deep learning-based methods—the GraphULM and CNN—demonstrate exceptional performance, achieving a perfect Jaccard index of 1. This indicates that the models refrained from making even a single false prediction throughout the length of the dataset. However, as the MB density increased, the overlap of MB became more pronounced, and thereby increasing the likelihood of false negatives. This can be reasoned well with, multiple overlapping PSFs appearing as a single broad PSF, complicating the localization process. Despite these challenges, GraphULM consistently outperformed all the other methods under consideration. The superior performance is largely due to the network's ability to learn contextual cues, particularly owing to the integration of graph networks.

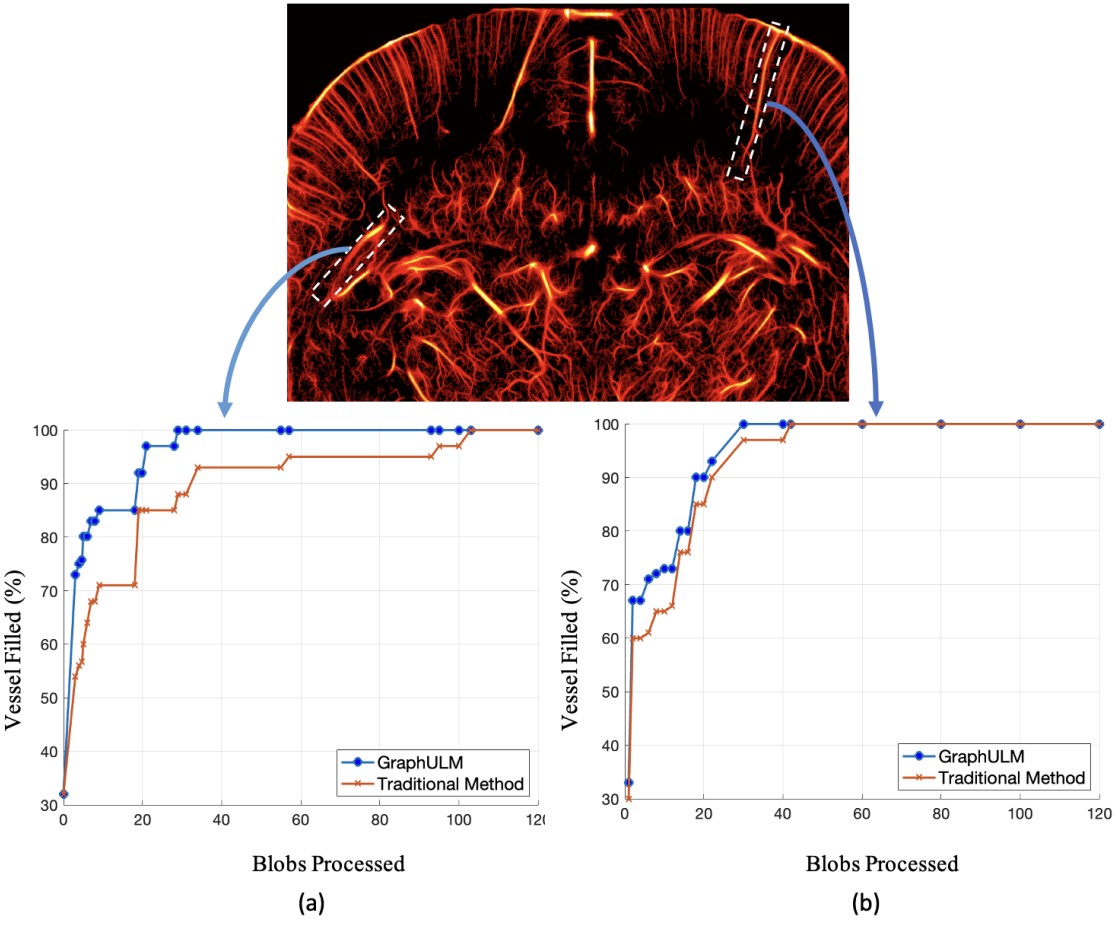

Figure 4: Evaluation of vessel filling accuracy for GraphULM and a traditional algorithm (Radial Symmetry). The top image displays the super-resolution vascular structure with highlighted regions, indicating the areas used for the analysis.

### 3.2. In-Vivo Evaluation

The in-vivo experiments were approved by the Institutional Committee (C2EA-59) (Heiles et al., 2022), and all imaging data used here is publicly available (Heiles et al., 2022). The dataset in (Heiles et al., 2022) involves two Sprague-Dawley rats; in one, continuous MB injection produced thousands of frames with an SNR of 29 dB. The data consisted of 800 frames per blob across 240 blobs and was acquired at a compounded frame rate of 1000 Hz using three tilted plane waves $[-3°, 0°, 3°]$ with a 15 MHz probe.

SVD filtering was applied to separate tissue from MB signals. The b-mode MB density ranged from 0.4–0.7 mm$^2$, consistent with our simulation setup. Each $78 \times 118$ frame was padded to $96 \times 128$ and divided into twelve $32 \times 32$ patches, each processed independently by GraphULM to predict a $256 \times 256$ high-resolution patch (upscaling factor 8). The patches were stitched to reconstruct the full image. White pixels were extracted via regional-max detection (with a constant threshold of 0.7 across all DL methods tested), rescaled, and used for tracking with a maximum linking distance of two pixels and a minimum trajectory length of 15 frames.

Figure 3 shows the super-resolution results and the associated metrics are shown in Table 1. As visible in Figure 3, the Graph-ULM method exhibits superior performance, particularly when evaluated visually. A higher number of bright pixels and a more comprehensive capture of vessels, especially in the top quadrant of the image (highlighted by the dashed-dotted green Box in Figure 3(a)), were observed. The white boxes in Fig. Figure 3 highlight the zoomed-in sections intended to draw attention to the specific details. In these areas, more veins are visible. In particular, Box-1 (Arrow-4) in Fig. 3(sub-(a))) shows a prominent strip (Highlighted by blue arrows in Figure 3(a)) of high pixel intensity, which appears absent in the images produced by other methods. This vein, extending from the left to the right corner of the image, represents one of the largest vessels in the in-vivo image with several smaller vessels branching from it. Given this structural prominence, it was anticipated that this vein would appear particularly bright throughout, and upon visual assessment, we observed that our method captured this feature most effectively. Furthermore, in Figure 3, the highlighted regions in Box-1 ( Arrow-3) and Box-2 (Arrow-2) reveal that the other methods introduce a degree of predictive fuzziness in densely vascular regions, whereas our approach effectively distinguishes closely spaced vessels. This serves as an indicator of fewer instances of false MB detection in the narrow gaps between vessels. In Figure 3 it is also worth noting, that in the regions represented by Box-1 (Arrows-1,3,5), Box-2 (Arrow-1), Box-3 (Arrows-1,2,3,4) and Box-4 (Arrow-2), our method identifies sharp and elongated vessels, whereas other methods tend to capture only segments or record less intensity distribution over the vessel. This advantage stems from the capability of our approach to visualize multiple image resolutions, as it employs three distinct filter sets, thus capturing intricate structural information more comprehensively.

Additionally, in the region highlighted by Figure 3(Box-2 (Arrow-4)) and Figure 3(Box-4 (Arrow-1)), the proposed method did not detect any vessels (represented as black spaces). Upon manual inspection of the periphery around these regions, we extrapolate that the absence of vessels in these regions appears to be the correct prediction. Further visual analysis of Figure 3 reveals additional insights; for instance, Locally Weighted Average method displayed noticeable gridding artifacts, which were absent in the other methods. Gridding

in CNN/LWA is likely due to pixel-shuffle aliasing and patch boundary interpolation issues, which GraphULM mitigates through global aggregation via GCN. Also, the Gaussian method produced a relatively fuzzy image with low visible contrast, whereas the Radial Symmetry approach exhibited substantial noise, particularly in the top quadrant. These observations indicate that deep learning-based methods, such as ours and CNN, are better suited for generating high-quality super-resolution images with minimal artifacts and enhanced clarity.

Table 1 presents the contrast scores for all methods discussed. The contrast score was evaluated by calculating the standard deviation of pixel intensity distribution in the in vivo images. Our method achieved a contrast score of 35.64, which was the highest among all evaluated methods.

To assess vessel filling percentage (Vfp), defined in Equation (5), we evaluated how quickly true MBs fill selected sharp vessels.

$$\text{Vfp} = \frac{\text{Number of MBs}}{\text{Total vessel area}} \times 100. \tag{5}$$

Figure 4(a)–(b) compares GraphULM with a traditional method, namely Radial Symmetry. GraphULM exhibits a steep initial rise, indicating earlier and more accurate MB detection, while traditional methods fill slowly because they favor high-intensity MB regions and often misplace centers near vessel boundaries. Both curves show similar peaks and plateaus since all methods estimate the approximate MB count reliably, even though traditional methods struggle with exact localization. Figure 5 presents the FRC using the first 50 blobs. GraphULM crosses the 1/2-bit threshold at the highest spatial frequency, corresponding to a resolution of 5.62 $\mu$m. Interestingly, despite its visible gridding, LWA performs well in FRC due to enhanced local contrast that preserves certain spatial frequencies. Gaussian outperforms radial Symmetry in FRC despite appearing fuzzier in Figure 3, reflecting that FRC measures frequency preservation rather than visual contrast. These results highlight the importance of objective metrics. Overall, GraphULM delivers strong localization accuracy, high resolution, and fast processing (Table 1), making it well suited for real-time ULM.

Table 1: The contrast score of the in-vivo image and the time to process one b-mode frame (in ms), normalized to the fastest implementation, for the five different methods are shown. GraphULM takes 10.8ms to process one patch of the b-mode image.

| Metric | GraphULM | CNN | LWA | GAU | RS |
|---|---|---|---|---|---|
| **Contrast** | **35.64** | 27.29 | 34.48 | 24.8 | 34.42 |
| **Time** | 1.16 | **1** | 3.16 | 5.40 | 3.56 |

## 4. Ablation

To validate the contribution of the GCN module to the overall architecture, we conducted an ablation study comparing GraphULM (with GCN) against the MRCNN-only variant

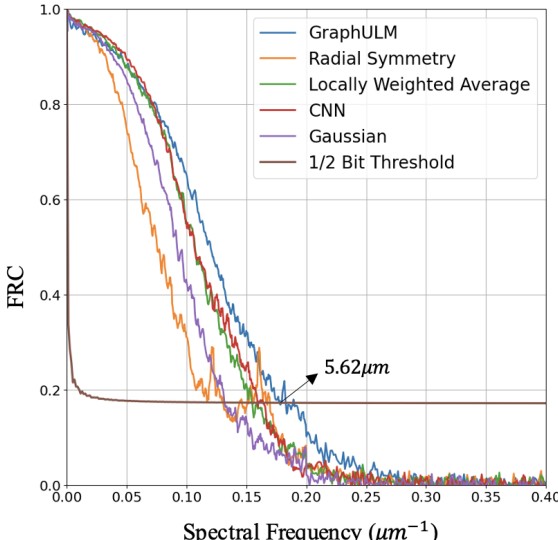

Figure 5: Frequency Ring Correlation (FRC) characteristics of various methods. The 1/2 bit threshold, represented by a horizontal line, denotes the spatial frequency at which the signal correlation falls to 0.5, indicating the resolution limit of each method. The intersection of each curve with this threshold indicates the resolution achieved, with GraphULM achieving a resolution of 5.62 µm.

(without GCN). We evaluated the performance on a synthetic dataset of closely spaced MB pairs at separation distances ranging from 5 to 22.5 µm, simulating challenging high-density scenarios where PSF overlap is significant. The synthetic dataset was generated with a pixel size of 1.25 µm (10 µm input pixel size with $8\times$ upsampling to $256 \times 256$ output resolution).

Figure 6 presents the detection accuracy results. At separations below 19 µm, the MRCNN-only variant consistently fails to resolve both bubbles, detecting only a single bubble due to its local receptive field struggling to distinguish overlapping PSFs without a global spatial context. In contrast, GraphULM successfully localizes both bubbles starting from a 10 µm separation. Notably, even at separations of 19 and 22.5 µm, where both methods were successful in making the prediction, GraphULM achieved 2 µm and 4 µm better localization accuracy, respectively, compared to the MRCNN-only variant. The graph structure naturally captures asymmetric MB distributions and contextual relationships, which is particularly valuable in high-density scenarios in which local CNN features alone are insufficient. These results provide empirical evidence that the GCN module contributes measurably to localization performance, with MRCNN's multi-scale feature extraction and GCN's global structural reasoning.

To provide an additional comparison with traditional feature detection methods, we evaluated the SIFT (Lowe, 2004) (Scale-Invariant Feature Transform) detector, which employs the Laplacian of Gaussian (LoG) operator to identify bubble locations and scales. The SIFT detector is particularly relevant for bubble detection because the LoG operator $((\nabla^2 G))$ is related to the physical definition of bubbles via Laplace pressure, making it a

theoretically motivated choice for this application. We considered a set of different sigma values (ranging from 0.3 to 0.7 $\lambda$) to determine the optimal scale-space extrema for peak detection and dark blob identification, following the multi-scale approach inherent to SIFT. We evaluated the performance on the InSilico flow dataset (Heiles et al., 2022), and to keep the comparison fair, we did not fine-tune our model on the PSF quality in the silico dataset. We believe that this helps attain an unbiased perspective on raw localization prediction. For tracking, we utilized the Hungarian algorithm, and both methods were tested on the same parametric set, with minimum gap closing in pairing set to 0 and minimum length of tracks set to 20.

Figure 7 shows the localization prediction results, and it is visually clear that the prediction of GraphULM appears to show high intensity in major portions of the silico flow image. Moreover, the thickness of all letters is more pronounced for GraphULM, indicating more complete vessel coverage and higher localization density. It is worth noting that many noisy tracks are visible in the SIFT intensity image, particularly near the bottom of the letter 'U' and the letter 'L', where numerous sparse, low-intensity detections (appearing as thin red lines ) form a background noise pattern. In contrast, the corresponding regions in the intensity image of GraphULM do not show these artifacts, which is indicative of fewer false positives and better noise suppression. The overall intensity accumulation in GraphULM was substantially higher, with peak counts reaching the upper range of the colorbar, while SIFT's maximum intensities of SIFT remained in the mid-range. The comparison highlights that while traditional methods like SIFT provide physically motivated and interpretable detection mechanisms, our approach GraphULM can learn more sophisticated feature representations that adapt to the specific characteristics of ULM data, resulting in improved localization accuracy, robustness to noise, and better handling of high-density scenarios with overlapping PSFs.

## 5. Conclusion

In this work, we propose a novel supervised architecture that combines an MRCNN and a GCN to process and localize MBs in ultrasound B-mode images. The pipeline is designed to leverage attention-like mechanisms for effective feature extraction. The inclusion of graphs in our model is pivotal, as they inherently capture global, learnable cues, thereby facilitating efficient learning. Concurrently, the MRCNN component, which employs multiple filter sizes in parallel, enhances the model's capacity to learn rich feature representations by incorporating diverse receptive fields. Additionally, the strategy of combining synthetic data with in-vivo samples for training, coupled with the need for fewer preset hyperparameters, has improved the model's robustness and generalization capability. To evaluate the effectiveness of our approach, we conducted extensive experiments and compared it with existing methods. In the FRC experiment, our method achieved a resolution of up to 5.62 $\mu$m. The combination of high localization accuracy and computational efficiency positions our approach as a promising candidate for real-time ULM applications. The method currently relies on postprocessing threshold values to achieve subwavelength localization. To further enhance the pipeline's speed, future work could focus on modifying the loss function to address joint optimization of localization accuracy and prediction probability.

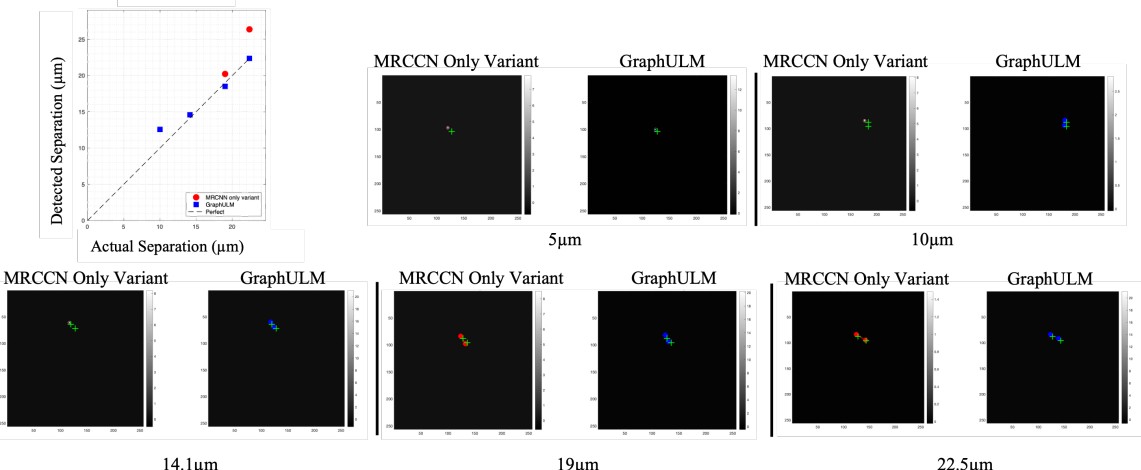

Figure 6: Ablation study comparing GraphULM with the MRCNN-only variant. (Top left) Scatter plot showing the detected versus actual separation distances. The dashed line represents the perfect detection. The visual detection results at actual separations of 5, 10, 14.1, 19, and 22.5 µm are shown in the image pairs. The green crosses indicate the ground truth positions, and the blue and red circles show the predictions. The MRCNN-only variant fails to resolve both bubbles distinctly until the separation distance becomes 19 µm.

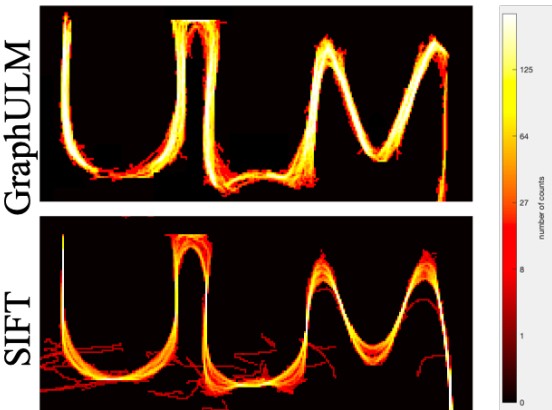

Figure 7: Comparison of GraphULM and SIFT-based detection on the InSilico flow dataset. Intensity images showing the super-resolution reconstruction of the "ULM" pattern. (Top) GraphULM demonstrates a tight, focused distribution of counts along the vessel structures with minimal background noise. (Bottom) SIFT shows a broader distribution with scattered low-intensity detections.

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

## Appendix A. Methods Compared

### A.1. Gaussain Fit

Gaussian fitting is know for its commendable performance though it comes with relatively high computational costs. It involves modeling the intensity profile of a MB as a Gaussian distribution. This methods considers a small region of interest around the MB and fits a Gaussian function to it. The center of the MB becomes the mean and the intensity spread corresponds to the standard deviation of the Gaussian. This mathematical fitting helps us explore more MBs in the image, subsequently it also helps in filtering out the noise. More details can be found in (Heiles et al., 2022).

### A.2. Radial Symmetry

The Radial Symmetry (RS) (Heiles et al., 2022) method takes advantage of the symmetrical nature of the intensity distribution around a MB. The gradient change and intensity variations around the MB are analyzed, to predict the radial center of the bubble. The center is called radial, since the intensity profile is explored in a circular pattern. This approach is particularly effective for sub-pixel localization because it does not rely on fitting a specific model but rather on the inherent symmetrical properties of the MB. RS has also been used for particle tracking in ULM.

### A.3. Locally Weighted Average

In the Locally Weighted Average (LWA) (Heiles et al., 2022) method, the position of MB is determined by computing the weighted average of intensity values within a region of interest. It is apparent that the brighter pixels will have a greater influence on the calculated center, since they will contribute significantly towards the computation performed. Overall this method is relatively simple yet effective for achieving sub-pixel localization and just like RS, it stays independent of the application of any model. LWA is particularly useful in scenarios where the MB shapes are irregular or affected by noise.

### A.4. CNN (mSPCN-ULM)

This method utilizes a Modified Sub-Pixel CNN (Liu et al., 2020) to localize MBs within ultrasound b-mode images, achieving a high spatial resolution with rapid processing speeds. The model is comprised with a total of 13 convolutional layers with intermittently employed residual connections, which are responsible for strengthening the feature propagation and network stability. This approach demonstrates high localization precision, even under conditions of dense MB distributions. This method is referred to as the CNN method throughout the paper.

