# OpenReview forum: "GraphULM: A Multi-Resolution CNN and GCN Framework for Ultrasound Localization Microscopy"
_MIDL.io/2026/Conference — MIDL 2026 Poster_

### Official Review · Reviewer_HsVz · 2026-01-08

**Confidence:** 3
**Preliminary Rating:** 3
**Final Rating:** 4

**Summary:**

The paper proposes a hybrid network for ultrasound localization microscopy, combining a multi-resolution CNN with a graph network. To improve microbubble localization accuracy, they model spatial relationships since it is difficult in high-density conditions. They train on synthetic data combining it with in-vivo frames, and also evaluate on in-vivo dataset showing an improved localization accuracy and qualitative vessel continuity compared to a CNN-based approach.

**Strengths:**

Clear motivation and presentation of the problem, the introduction of spatial reasoning makes sense for this localization problem due to microbubble density.
Clear architecture design. The results show to outperform the baseline CNN and other models, also visible in the qualitative images with details.

**Weaknesses:**

The paper claims that combining MRCNN with graphs improves the results, but they lack an ablation study to show the performance of MRCNN alone and why addition of graph reasoning is meaningful. The MRCNN alone is also unclear, it is not introduced as a novel model or referenced.
It is also unclear what in-vivo data they have used for the training. Is it part of the evaluation data, if so, is there an overlap in acquisition consitions?

**Detailed Comments:**

Is MRCNN a novel idea and a contribution, or what is the reasoning to use it?

**Justification Of Final Rating:**

The authors mainly added ablation studies comparing the MRCNN-only variant with the GCN on a synthetic dataset and feature detection of flow image. Given the very small distances, the GraphULM seems to be better at separation of MB pairs. And the flow image results also show more consistent and localized feature detection.

**Justification Of The Preliminary Rating:**

The paper presents an interesting and well motivated application, using graph based reasoning in medical images is not a novel idea but to my knowledge it is new in this specific application. The lack of ablation stidues and unclear usage of MRCNN as backbone limit the methodological claims of the paper.

**Questions To Address In The Rebuttal:**

Mainly addressing the issues mentioned in weaknesses

---

### Official Review · Reviewer_GS1h · 2026-01-08

**Confidence:** 3
**Preliminary Rating:** 3
**Final Rating:** 3

**Summary:**

The paper introduces a supervised method combining convolution and graph NNs to  localize microbubbles in Ultrasound Localization Microscopy. The method is trained and tested on a synthetically generated data and an in-vivo dataset. Qualitative and quantitative results are provided including a Frequency Ring Correlation.

**Strengths:**

The paper provides a clear and well-organised description of the proposed method, which makes it easy to follow the technical contributions and underlying ideas. The manuscript is generally well written, with a coherent structure and good flow, enabling the reader to understand both the motivation and the methodological details without ambiguity.

**Weaknesses:**

While the method is clearly presented, the rationale for combining a CNN with a graph-based network is not sufficiently justified. The paper would benefit from a deeper explanation of how the two components complement each other, and why this hybrid design is preferable to either architecture alone. Moreover, the absence of an ablation study makes it difficult to assess the contribution of each component and to understand whether the graph module provides measurable added value. In addition, the manuscript lacks a clear description of competing or baseline methods used for comparison, which limits the reader’s ability to contextualise the reported performance. Without these details, it becomes challenging to evaluate the true significance and impact of the proposed approach.

**Detailed Comments:**

•	Abstract: “The process of localization, followed by tracking” the subject is missing in this sentence. What do you localise and track? I suppose microbubbles.
•	Section 2: “Hungarian algorithm” add reference.
•	Add a section 2.4 on method comparisons, describing the competing methods reported in Figure 2: (b) CNN, (c) Locally Weighted Average, (d) Gaussian, and (e) Radial Symmetry.
•	In Figure 2, GraphULM and CNN names are reported, while in the text, they are called GCN and MRCNN. I guess GraphULM is the entire method, while CNN is a competing method? It would be important to have both GCN and MRCNN alone in Figure 2.

**Justification Of Final Rating:**

The rebuttal did not sufficiently address the main concerns raised in the review. The ablation study remains incomplete, and Figure 6 does not provide meaningful insight into the contribution of the individual model components, particularly in the summary results. Overall, the paper did not improve substantially after the rebuttal, and previously identified issues persist, including the lack of clear justification for the hybrid CNN–graph architecture and the insufficient description of comparison methods. As a result, the core questions regarding the methodological contribution and empirical validation remain unresolved.

**Justification Of The Preliminary Rating:**

The paper is clearly written and presents an interesting idea, but the empirical validation is incomplete. Without ablations or clear baselines, it is difficult to assess the real contribution of the method, leading to a borderline recommendation.

**Questions To Address In The Rebuttal:**

The authors should clarify the motivation for combining CNN and graph-based modules and provide evidence that each contributes to performance. It would also help to specify the baseline methods used for comparison and justify their selection. An ablation or component-wise analysis would substantially strengthen the empirical validation of the approach.

---

> ### Comment · Area_Chair_Kwc3 · 2026-02-01
> **Final rating**
>
> Dear reviewer,
>
> Could you please provide your final rating for this submission?
>
> Thank you!

---

### Official Review · Reviewer_6X1g · 2026-01-10

**Confidence:** 4
**Preliminary Rating:** 3
**Final Rating:** 4

**Summary:**

Interesting framework attempting to identify microbubbles in images, multi-resolution CNN, GCN networks,
  um range resolution is interesting.
 Other comments below.
 Other comments below.
 Other comments below.
 Other comments below.
 Other comments below.
 Other comments below.
 Other comments below.

**Strengths:**

The theme of identifying maxima of microbubble densities as in Figure 1. is interesting.
The simulation is great.
The microbubble domain is niche but interesting, particularly to obtain the image resolution.

**Weaknesses:**

Figure 1 looks essentially like peak detection. There has been previous work identifying image bubble structures in ultrasound, at the mm range.
The work here presents "radial symmetry" as "representative of all traditional methods", it's not clear what or how radial symmetry is evaluated. In terms of traditional methods, couldn't the SIFT feature detector be incorporated as a detector? It would be good to mention something along these lines. It's been previously used to detect bubbles in B-mode ultrasound of the brain.

[a] Machado, Inês, et al. "Non-rigid registration of 3D ultrasound for neurosurgery using automatic feature detection and matching." International journal of computer assisted radiology and surgery 13.10 (2018): 1525-1538.
https://pmc.ncbi.nlm.nih.gov/articles/PMC6151276/

[b] Toews M, Wells WM. Phantomless Auto-Calibration and Online Calibration Assessment for a Tracked Freehand 2-D Ultrasound Probe. IEEE Trans Med Imaging. 2018 Jan;37(1):262-272. doi: 10.1109/TMI.2017.2750978. Epub 2017 Sep 11. PMID: 28910761; PMCID: PMC5808952.

Bubbles are defined physically in 3D by the Laplace pressure
https://en.wikipedia.org/wiki/Laplace_pressure

The SIFT feature detector maximizes the Laplacian ∇^2 of Gaussian operator to identify bubble location and scale.

Figure 3: what are we looking at here? Are these brain images, how were they acqired? There seems to be no description.

**Detailed Comments:**

The approach looks interesting, the domain (ultrasound microbubbles) is promising but quite narrow.

It seems the approach might be simplified and/or improved by incorporating traditional bubble detectors, e.g. SIFT.

A bubble is defined by the Laplace pressure equation, it would be good to see a reference to the bubble equation.
https://en.wikipedia.org/wiki/Laplace_pressure

**Justification Of Final Rating:**

With the added discussion of peak detection and SIFT, I raise my score to weak accept.
************************************************************************************************************************************

**Justification Of The Preliminary Rating:**

A novel architecture in a niche imaging domain. Not enough comparison vis a vis traditional detector baseline.
A novel architecture in a niche imaging domain. Not enough comparison vis a vis traditional detector baseline.
A novel architecture in a niche imaging domain. Not enough comparison vis a vis traditional detector baseline.

**Questions To Address In The Rebuttal:**

Address the above.

---

> ### Comment · Area_Chair_Kwc3 · 2026-02-01
> **Final rating**
>
> Dear reviewer,
>
> Could you please provide your final rating for this submission?
>
> Thank you!

---

### Author Rebuttal · Authors · 2026-01-25

**Rebuttal:**

We thank the reviewers for their constructive feedback and thoughtful evaluation. Changes in the revised manuscript are made in blue and below we address each comment concisely.

@Reviewer 1 (6X1g)
Regarding Fig.1, we agree it can resemble peak detection. After clutter removal, MBs appear as dominant PSF-like intensity patterns in B-mode images, and ULM aims to localize the PSF center.
For traditional baselines, Radial Symmetry was used as a representative method in the vessel-filling experiment because it produced the best visual in-vivo result among evaluated classical approaches.  Simulations showed that all traditional methods exhibit highly overlapping performance curves. For clarity, we therefore present one representative method in the main text and report results across all traditional baselines in an appendix.
Following the valuable recommendation, we added an ablation comparing GraphULM with SIFT-based detection on the InSilico dataset (Fig.7). GraphULM consistently achieves improved localization, supported by added ablation. Invivo dataset details and inference/tracking procedures for Fig. 3 are clarified in Sec. 3.2.

@Reviewer 2 (GS1h)
We thank the reviewer for this valuable feedback. A comprehensive ablation study (Fig's 6 and 7) now directly addresses why graph modules add measurable value. In particular, one experiment compares GraphULM against the MRCNN-only variant (baseline) on closely spaced MB pairs with separation distances ranging from 5 to ~23 micro-m. Furthermore, section 2.2 was expanded to clarify the rationale for combining MRCNN and GCN. The appendix section now explains all methods. Also, we have competed GraphULM with CNN, not with GCN.

@Reviewer 3 (HsVz)
We thank the reviewer for the positive assessment and for the constructive feedback. We have addressed the concerns by adding a comprehensive ablation study (Figures 6 and 7) that directly compares GraphULM with the MRCNN-only variant. This ablation isolates the contribution of the GCN component and helps clarify its role within the proposed architecture. We clarify that the MRCNN is not a novel contribution but a standard backbone choice for multi-scale feature extraction. The novelty lies in integrating it with a GCN for ULM localization, which is now explicitly stated in sec 2.2. The authors would like to clarify, that the in-vivo frames used for training were obtained from a separate acquisition (uninferred blobs) and do not overlap with the evaluation dataset.

**Supporting Material:**

/attachment/7d94dbd6e4b2de5f6718236447d35f8ffe9304ce.pdf

---

### Meta-Review · Area_Chair_Kwc3 · 2026-02-07

**Recommendation:** Reject
**Confidence:** 4

**Metareview:**

The paper proposes a deep learning approach for ultrasound localization microscopy (ULM) based on B‑mode ultrasound images. Prior work has demonstrated that CNNs can effectively localize microbubbles in this setting. The present submission extends such CNN‑based approaches by introducing an additional graph convolutional network (GCN) branch. The model processes each image through a multi‑resolution CNN (MRCNN) pathway and a GCN pathway in parallel, after which the GCN output modulates the CNN features. The authors argue that this GCN branch provides “global structural context” beyond what the CNN can capture.

Two of the three reviewers raised substantial concerns regarding both the justification for introducing the GCN branch and the lack of supporting ablation studies. Although the authors added a limited ablation in the revised version, I concur with reviewer GS1h that the most important issues remain insufficiently addressed.

The main weakness of the submission is that its central contribution – the inclusion of a GCN branch – is neither well‑motivated nor convincingly validated. The added ablation study examines only a very narrow synthetic scenario and does not integrate the MRCNN‑only baseline into the broader set of experiments. Moreover, important variants are missing, such as a GCN‑only model, versions with deeper GCNs to examine local vs. global effects, or statistical analyses of performance differences. As a result, the evidence presented does not allow the reader to assess whether the GCN contributes meaningfully to model performance.

Most critically, the authors’ conceptual interpretation of their GCN component appears fundamentally flawed. The paper repeatedly claims that the GCN captures global structural context. However, based on the architecture described, the GCN consists of two one‑hop message‑passing layers and therefore has an intrinsically local receptive field. This design cannot provide the global reasoning the authors attribute to it. In fact, the multi‑resolution CNN pathway, with several stacked convolutional blocks and residual connections, likely has a substantially larger effective receptive field than the GCN. Consequently, the statements regarding complementary roles - CNN for local details and GCN for global context - are not supported by the model design.

Given these issues, the paper’s central claim remains unsubstantiated, and the empirical evidence is not yet sufficient to support acceptance.

---

### Decision · Program_Chairs · 2026-02-13

Accept (Poster)